# Prevalence of active tuberculosis disease among healthcare workers and support staff in healthcare settings of the Amhara region, Ethiopia

**Melashu Balew Shiferaw** [1] *, **Mulusew Alemneh Sinishaw** [2], **Desalegne Amare**[3], **Genetu Alem**[4], **Dawit Asefa**[5], **Eveline Klinkenberg**[6,7]

1 Research and Technology Transfer Directorate, Amhara Public Health Institute, Bahir Dar, Ethiopia, 2 Department of Clinical Chemistry, College of Medicine and Health Sciences, Bahir Dar University, Bahir Dar, Ethiopia, 3 School of Public Health, College of Medicine and Health Sciences, Bahir Dar University, Bahir Dar, Ethiopia, 4 Amhara Regional Health Bureau, Bahir Dar, Ethiopia, 5 KNCV Tuberculosis Foundation/USAID Challenge TB, Addis Ababa, Ethiopia, 6 KNCV Tuberculosis Foundation, The Hague, The Netherlands, 7 Department of Global Health, Amsterdam Institute for Global Health and Development, Amsterdam University Medical Centers, Amsterdam, The Netherlands

* bmelashu@gmail.com

**Data Availability Statement:** All relevant data are within the manuscript.

## Abstract

### Background

Health care workers (HCWs) are at an increased risk of acquiring tuberculosis (TB) compared to the general population, because of the frequent face to face contact or potential exposure to TB through shared air or space with infectious patient(s), regardless of economic setting and local TB incidence. Information on the burden of active TB disease among HCWs will help guide control measures, can be utilized to evaluate the effectiveness of TB infection prevention programs in the health care setting, and guide necessary actions. However, data on TB among HCW in Ethiopia is limited. Therefore, this study aimed to determine the prevalence of active TB disease among healthcare workers and support staff in healthcare settings in the Amhara region of Ethiopia.

### Methods

A cross-sectional study design was used to recruit a total of 580 randomly selected study participants in the Amhara region. Data were collected over four months in selected hospitals and health centers. Implementation of TB prevention and control measures was evaluated using a standardized checklist. The main outcome indicator was active TB as measured by a laboratory diagnosis using GeneXpert technology.

### Results

A total of 580 study participants were enrolled. The mean age was 31.3 (±7.8 standard deviation) years, with about two-thirds (65.3%) aged between 18–24 years. A total of 9 (1.6%) MTB cases were detected, 4 (1.4%) in HCWs and 5 (1.7%) in support staff, which did not significantly differ (P = 0.50). About 90% of the participants had not received TB infection

**Funding:** MBS received the award. The Global Health Bureau, Office of Infectious Disease, US Agency for International Development, financially supported this study through Challenge TB under the terms of Cooperative Agreement No. AID-OAA-A-14-00029. This study is made possible by the generous support of the American people through the United States Agency for International Development (USAID) / Challenge TB project. The funders had no role in study design, data collection and analysis, decision to publish, or preparation of the manuscript. The contents are the responsibility of the authors and do not necessarily reflect the views of USAID or the United States Government.

**Competing interests:** The authors have declared that no competing interests exist.

prevention and control training ever. More than half (54%) of the study participants worked in poorly ventilated rooms. Triage of coughing patients was not practiced in 32% of the studied facilities (health centers and hospitals).

## Conclusions

The magnitude of TB among healthcare workers and support staff in healthcare settings was higher than in the general population (140 per 100000 population). The status of implementation of tuberculosis prevention and control measures indicated missed opportunities. Hence, strict implementation of developed infection control plans of TB in healthcare settings needs to be improved.

## Background

Tuberculosis (TB) is an infectious disease caused by *Mycobacterium tuberculosis* that most often affects the lungs [1]. Transmission of TB and multidrug-resistant TB (MDR-TB)/ extensively drug-resistant TB (XDR-TB) typically is more intense in congregates settings like in health facilities, laboratories, prisons, military camps, and others [2]. Health care workers (HCWs) are at an increased risk of acquiring TB compared to the general population, because of the frequent face to face contact or potential exposure to TB through shared air or space with infectious patient(s), regardless of economic setting and local TB incidence [3]. A systematic review of TB incidence in low- and middle-income countries estimated the annual risk of TB infection in HCWs to range from 3.9% to 14.3% [4] with HCWs having 1.9 to 5.7 times more risk of developing active TB depending on the country setting and the level of infection control measures in healthcare facilities [5].

Health care workers are also at greater risk for MDR-TB than the general population and are up to six times more likely to be hospitalized for MDR-TB [6]. With the global increasing incidence of MDR-TB and XDR-TB, the problem has been further compounded, with the risk of HCWs contracting more severe forms of the disease, which are difficult or sometimes impossible to treat successfully [7, 8].

Despite the higher risk, HCWs may be less likely to seek medical care leading to under-reporting of the true TB disease incidence [5]. Reported reasons for this are less attention is given by policymakers to health care workers TB status, negligence of healthcare workers in the adaptation of the TB working environment due to shortage of personal protective equipment (PPE) and lack of supervision, and fear of stigma and discrimination [9]. Delayed care seeking results in delayed diagnosis, risking less effective treatment for drug-resistant TB (DR-TB), while longer contact with patients increases the risk of ongoing transmission to HCWs [6, 10, 11]. In Ethiopia, data about TB disease in HCWs is limited. Knowing the prevalence of active TB among HCWs in the Ethiopian setting would help to determine the effectiveness of the TB infection prevention program and guide further control measures.

The WHO End TB Strategy, approved by the World Health Assembly in 2014, calls for a 95% reduction in TB deaths and 90% reduction in the TB incidence rate by 2035, compared to 2015 [12]. Infection prevention is a cost-effective method of reducing TB transmission. Implementing effective TB infection prevention and control measures can be accomplished in manageable steps and will contribute to the overall protection of the health workforce, the community and vulnerable populations, from air born transmissions. Besides the use of personal protective equipment (PPE), regular screening of HCWs for TB and other infections like

HIV is part of the routinely recommended package of preventive control measures for health care settings that could help reduce transmission together with effectively triaging people with TB symptoms upon arrival at the health facility, promoting good cough etiquette, diagnosing people with TB symptoms early and starting treatment without delay as well as adequate ventilation of the consulting and waiting rooms [13, 14]. Some studies conducted in Ethiopia revealed that practices of TB infection prevention and control measures were below the standard [15–17]. Therefore, this study aims to evaluate the implementation of TB infection prevention and control measures in addition to estimating the prevalence of TB in health care workers.

## Methods

### Study design and study area

A cross sectional study was conducted to determine the prevalence of active TB disease among healthcare workers in Amhara region. The study was carried out from October 01, 2018 to January 30, 2019. Amhara regional state is one the largest Federal States of Ethiopia and has 841 health centers and 67 hospitals that provide TB services. The region has 24 GeneXpert sites and a total of 150 so-called District Sample Collection Centers (DSCCs) that collect sputum specimens of presumptive MDR-TB patients from remote health facilities for testing at the two regional laboratories and one TB culture laboratory assigned to diagnose drug resistance TB. Nine treatment initiating centers (TIC) are providing treatment for MDR TB patients. There are also 115 treatment follow up centers (TFCs). In the 2009 Ethiopian fiscal year (08 July 2016 to 07 July 2017), the Amhara region reported 23,345 TB cases (all forms) and 4896 (21%) bacteriology confirmed TB cases. The prevalence of TB and HIV co-infection in the region was 7.3% (1625/22,206) in the same period. The Amhara region had a reported 29,801 health care workers (13,440 males and 16,361 females) in the 2009 Ethiopian fiscal year in addition to 14,296 administrative and support staff working in the health care setting [18].

### Study population

All healthcare workers and support staff recruited under the Amhara Regional State Health Bureau health facilities were our study population.

**Inclusion and exclusion criteria.**   Health care workers who worked in the Amhara region health facilities departments were included in the study. Both clinical, and administrative and support staff were included to assess the burden of TB among different groups. Since all facilities provided TB service, there was no exclusion of facilities in the sampling frame.

### Sampling

The sample size for this study was determined using single population proportion formula assuming a potential 9.0% TB prevalence among health care workers based on data from South Africa [10] taking into consideration similar practice and exposure for TB for health care workers with Ethiopia, and assuming a 95% confidence interval, 5% marginal error and a design effect of 2. With these assumptions the total sample size was 252.

By adding 15% contingency for non-response, the target sample size for each group (health care workers and support staff) was set at 290, assuming both to have a similar burden. Thus, the total target sample size for the study was 580 participants to be screened for TB. Support staff included cleaners, cashiers, data clerks, case managers who are assigned to help patients in guiding and facilitating the diagnosis and treatment in health facilities, card room workers, security guards and health facility assigned treatment supporters who are assigned to follow treatment adherence of patients using different communication mechanisms such as phone call).

Multi stage cluster sampling was used in order to select the participants for each of the two target groups. Stage 1, consisted of randomly selecting five zones out of the total of 13 administrative zones in Amhara region. At stage 2, we proportionally selected a number of HFs for each zone based on the total number of HFs in each zone. The average number of health facilities (HFs) in each selected zone was 52 in Awi, 12 in Bahir Dar, 102 in South Gondar, 109 in East Gojjam, and 70 in North Wollo. Even if the national recommendation determining 12 sites sampled are sufficient at regional level, we included a total of 38 sampled HFs to make the study more representative. Finally, at each site 8 HCWs and 8 support staffs were included, resulting in a total of 38 x 8 = 304 targeted staff per group. Selection of staff at each site was done using the type of health facilities and the type of services they provide. Participants at each selected health facility were included using systematic random sampling by getting the list of health care workers and support staff from the human resource department. Then, each selected staff was invited to participate in the study. When they had not have been assigned to work the day of the visit, an appointment was set to conduct the procedures another day.

## Data collection procedure and laboratory analysis

Trained supervisors and data collectors were recruited to conduct the data collection. Data were collected using five teams, each composed of two data collectors (one laboratory technician and one clinician) and a supervisor. They were recruited for the project and assigned to collect the data from recruited participants (HCWs and support staff) in the sampled health facilities.

A questionnaire was used to collect socio-demographic data and TB exposure to known TB patients for each participant. The status of TB infection prevention and control measures was assessed in each health facility and details about the use of PPE were inquired about in the questionnaire. The ventilation status of the room that the staff usually worked in was assessed by the data collectors during interviewing of the study participants. The room was evaluated and considered ventilated if a room had open window and cross. Others who did not have room to work in like security guards were not evaluated. All study participants were evaluated for TB symptoms by trained professional nurses/health officers. Based on the Ethiopian national TB screening algorithm, those who had no symptoms suggestive for active TB were considered TB screen negative and not to have TB. Participants who reported a cough of 2 weeks or more, or presence of fever, weight loss or night sweats with any duration were considered TB symptomatic. All symptomatic participants were asked to provide one sputum sample (on the spot). All sputum samples were processed by trained laboratory personnel at GeneXpert sites using the GeneXpert technology (Xpert ®MTB/RIF) for the diagnosis of TB. If the facility itself did not have a machine, collected specimens were transported at 2–8 $^0$C to the nearest diagnostic sites using triple packaging within three days of sputum collection in order to minimize transportation frequency and cost.

## Data management, analysis and quality assurance

During data collection, questionnaires were checked daily for accuracy, completeness, and consistency by the supervisors. Epi-info version 7.2.2 Software was used for data entry and SPSS version 20 statistical package for the data analysis. Frequencies and cross tabulations were used to describe study participants. The main outcome was presence of active TB (MTB detected/ Rif resistant or susceptible) within the group investigated.

To enhance data quality a structured questionnaire was used. Two days training was given for the data collectors on the data collection tools and field work methods. The investigators strictly followed and made on site supervision during the whole period of data collection.

Laboratory diagnosis was done by trained laboratory personnel and the testing was done at diagnostic sites. Invalid/error results were repeated. To ensure data quality and avoid transcription errors, double data entry was done for all the questionnaires. Each laboratory result recorded was cross checked with the GeneXpert machine during data entry.

### Ethical consideration

This study was reviewed and approved by the Amhara Public Health Institute Ethical Review Board. Support letters were obtained from the Amhara regional health bureau, zonal health departments, Woreda health offices and health facilities. Written informed consent was obtained from each study participant and participation was voluntary. Only code number was used to ensure confidentiality. Result of TB diagnosis was communicated only in a private and secured manner. All the diagnosed TB cases were linked to care.

## Results

### Socio-demographic characteristics of study participants

A total of 580 study subjects, 290 HCWs and 290 support staff, were enrolled from 38 health facilities in this survey. The mean (±SD) age of the participants was 31.3 (±7.8) years, with two thirds (379[65.3%]) being between 18–24 years. The male/female ratio differed significantly between the two groups, with 112 [38.6%]) of the HCW workers being female, and 190 (65.5%) of the support staff being female (P<0.001). Among the HCW, most of the participants were nurses (21.0%) with the rest being composed of laboratory staff (8.6%), pharmacy staff (5.9%), public health officers (4.8%), radiographers (4.1%), and physicians (3.6%) in profession. A total of 179 (61.7%) support staff and 135 (46.6%) HCWs had work experience shorter than five years (Table 1).

### TB exposure status and practice of infection prevention and control measures

Of the HCW, 28% had experience of working in the TB clinic. The majority of the HCWs (82.8%) and support staff (89.3%) reported not having a BCG vaccination. None of the study participants were screened for TB as part of entry screening when they started working in the health facility. A total of 538 (92.8%) of all participants indicated they were not annually screened for TB as part of the facility TB infection prevention screening program, this did not differ by group (P = 0.423). The majority of the HCWs (237 [81.7%]) and support staff (284 [97.9%]) were not trained on TB infection control. Three quarter (75.0%) of HCWs reported having had direct contact with TB patients during their work in the health facility during the previous year. Moreover, 147 (50.7%) HCWs and 60 (65.9%) support staff worked in poorly ventilated rooms. In this study, 32 (11.0%) support staff and 1 (0.3%) HCW reported having ever taken isoniazid TB preventive therapy (Table 2). The majority of the healthcare workers, 456 (78.7%), did not have nor use N95 respirators as a personal protection measure. Of the 124 participants who had N95 respirator access, 82 (66.1%) did not use the respirator at work. Only 26.3% (10/38) of the health facilities had a TB IC plan. Of which, 70% (7/10) indicated they did not audit their plan annually as recommended.

### TB screening and disease status

A total of 103 (17.8%) participants, 46 (15.9%) HCWs and 57 (19.7%) support staff, were identified as presumptive TB based on reported symptoms. Cough of two weeks or more was reported by 40 (13.8) HCWs and 37 (12.8) support staff. Weight loss, night sweats and fever

**Table 1. Socio-demographic characteristics study participants, Amhara region.**

| Characteristics | Category | Participants | | |
| --- | --- | --- | --- | --- |
| | | Support staff | HCWs | Total |
| **Sex** | Male | 100(34.5) | 178(61.4) | 278(47.9) |
| | Female | 190(65.5) | 112(38.6) | 302(52.1) |
| **Age in years** | 18–24 | 42(14.5) | 26(9.0) | 68(11.7) |
| | 25–34 | 168(57.9) | 211(72.8) | 379(65.4) |
| | 35–44 | 49(16.9) | 34(11.7) | 83(14.3) |
| | >44 | 31(10.7) | 19(6.5) | 50(8.6) |
| | Mean(±SD) | 32.0(±8.8) | 30.7(±6.5) | 31.3 (±7.8) |
| **Job title** | Cleaner | 58(20.0) | - | 58(10.0) |
| | Data clerk | 51(17.6) | - | 51(8.8) |
| | Casher | 42(14.5) | - | 42(7.2) |
| | Security guard | 34(11.7) | - | 34(5.9) |
| | Porter | 30(10.3) | - | 30(5.2) |
| | Case manager | 26(9.0) | - | 26(4.5) |
| | Treatment supporter | 18(6.2) | - | 18(3.1) |
| | Card room | 17(5.9) | - | 17(2.9) |
| | Receptionist | 14(4.8) | - | 14(2.4) |
| | Nurse | - | 122(42.1) | 122(21.0) |
| | Laboratory staff | - | 50(17.2) | 50(8.6) |
| | Pharmacy staff | - | 34(11.7) | 34(5.9) |
| | Public health officer | - | 28(9.7) | 28(4.8) |
| | Radiographers | - | 24(8.3) | 24(4.2) |
| | Physician | - | 21(7.2) | 21(3.6) |
| | Midwife | - | 11(3.8) | 11(1.9) |
| **Work experience** | <5 years | 179(61.7) | 135(46.6) | 314(54.2) |
| | 5–10 years | 72(24.8) | 112(38.6) | 184(31.7) |
| | >10 years | 39(13.5) | 43(14.8) | 82(14.1) |

Case manager: support staff assigned to help patients in guiding and facilitating the diagnosis and treatment in health facilities; Treatment supporter: support staff assigned to follow treatment adherence of patients using different communication mechanisms such as phone call; HCWs: healthcare workers.

were reported by 42 (7.2%), 23 (4.0%) and 35 (6.0%) of participants, respectively. Active tuberculosis was found in 9 (1.6%) participants: 4 (1.4%) HCWs and 5 (1.7%) support staff, and there was no difference between these groups (P value: 0.50). Out of the 9, five reported to be currently on the intensive phase of TB treatment for bacteriological confirmed TB. Forty (6.9%) participants reported a previous history of TB disease, and only one of those diagnosed with TB in the study reported a previous episode of TB (Tables 3 and 4). The identified cases belonged to the following cadres of staff, two case managers who were support staff assigned to help patients in guiding and facilitating the diagnosis and treatment in the health facilities, one cleaner, one cashier, two pharmacists, one physician, one laboratory technologist and one security guard. None of the identified cases had drug resistant TB.

## Discussion

In this study, 1.4% of HCWs and 1.7% of support staff in selected facilities in the Amhara region, Ethiopia had active tuberculosis. The selected facilities were representative for the region. Implementation of IC measures was low though numbers included in the study were limited. The observed prevalence (1.6% ~ 1600 per 100,000 population), though based on a

**Table 2. TB exposure status of study participants in Amhara region.**

| Characteristics | Category | Participants | | | |
|---|---|---|---|---|---|
| | | Support staff | HCWs | Total | P value |
| Has experience of TB patient care in TB clinic | No | NA | 210(72.4) | 210(72.4) | - |
| | Yes | NA | 80(27.6) | 80(27.6) | - |
| BCG vaccinated | No | 259(89.3) | 240(82.8) | 499(86.0) | 0.023 |
| | Yes | 31(10.7) | 50(17.2) | 81(14.0) | |
| Trained on TB infection control | No | 284(97.9) | 237(81.7) | 521(89.8) | 0.0001 |
| | Yes | 6(2.1) | 53(18.3) | 59(10.2) | |
| Screened for TB as part of a facility screening program | No | 266(91.7) | 272(93.8) | 538(92.8) | 0.423 |
| | Yes | 24(8.3) | 18(6.2) | 42(7.2) | |
| Has worked on triaging for coughing patients | No | 278(95.9) | 140(48.3) | 418(72.1) | - |
| | Yes | 12(4.1) | 150(51.7) | 162(27.9) | |
| Having direct contact with TB patients in the last year | Yes, own house | 7(2.4) | 6(2.1) | 13(2.2) | - |
| | Yes, in the health facility | 174(60.0) | 219(75.5) | 393(67.8) | |
| | Yes, outside household | 0(0.0) | 2(0.7) | 2(0.3) | |
| | No | 91(31.4) | 58(20.0) | 149(25.7) | |
| | Do not know | 18(6.2) | 5(1.7) | 23(4.0) | |
| Ever use medication such as isoniazid to prevent TB | Yes | 32(11.0) | 1(0.3) | 33(5.7) | - |
| | No | 258(88. 9) | 289(99.7) | 547(94.3) | |
| Ventilation status of the room as observed by data collectors | Not well ventilated | 60(65.9) | 147(50.7) | 207(54.3) | 0.011 |
| | Well ventilated | 31(34.1) | 143(49.3) | 174(45.7) | |
| Room cleanness as observed by data collectors | Yes | 165(72.1) | 268 (82.4) | 433(83. 5) | - |
| | No | 64(27.9) | 22(7.6) | 86(16.6) | |

NA: not applicable

small sample is significantly higher than the national TB prevalence for the general population, estimated at 140 per 100,000 populations [19]. A recent study conducted in Zambia showed at least 0.5% of HCWs screened had TB during the screening appointment [20], which is lower compared to our findings. The higher prevalence of TB in HCWs and support staff suggests an occupational hazard, possible linked to poor status of TB prevention and control measures in healthcare settings including our setting. This needs an effective implementation of TB infection control (TBIC) to reduce the TB burden, ongoing transmission and their consequences in health facilities [20, 21]. Our finding is lower than a study in Kenya that reported 3.0% prevalence of active TB among healthcare workers [22]. This difference might be due to differences in inclusion criteria as the Kenyan study only included symptomatic cases diagnosed by microscopy and reviewed retrospectively from the laboratory register.

Routine TB screening for all staff working in health care facilities is important to identify TB cases timely. Our data indicated that over 90% of the study participants were not regularly screened for TB. However, the national TB guideline recommend all healthcare settings to conduct TB screening for all staff before entry into the facility as frequent rotation of staff which would pose more transmission risk so screening is extra important, and encourages periodic TB screening in facilities [23].

Triage of people with TB signs and symptoms, or with TB disease, is recommended to reduce *M. tuberculosis* transmission to health workers, persons attending health care facilities or other persons in settings with a high risk of transmission [23, 24]. The effective implementation of triage goes beyond the minimal infrastructure requirements. It should also include fast tracking of patients with presumed TB, rapid diagnosis, and respiratory separation, use of

**Table 3. TB diseases status and symptoms in Amhara region.**

| Characteristics | Category | Participants | | |
|---|---|---|---|---|
| | | Support staff | HCWs | Total |
| **Active TB** | Negative | 285(98.3) | 286(98.6) | 571(98.4) |
| | Positive/sensitive | 5(1.7) | 4(1.4) | 9(1.6) |
| | Positive /resistance | 0 | 0 | 0 |
| **Ever have TB diseases** | Yes currently on treatment | 3(1.0) | 2(0.7) | 5(0.9) |
| | Yes previous | 30(10.3) | 10(3.4) | 40(6.9) |
| | No | 245(84.5) | 276(95.2) | 521(89.8) |
| | Do not know | 12(4.1) | 2(0.7) | 14(2.4) |
| **TB screenin** | Positive | 57(19.7) | 46(15.9) | 103(17.8) |
| | Negative | 233(80.3) | 244(84.1) | 477(82.2) |
| **Cough** | No | 253(87.2) | 250(86.2) | 503(86.7) |
| | Yes | 37(12.8) | 40(13.8) | 77(13.3) |
| **Cough of two weeks or more** | No | 263(90.7) | 262(90.3) | 525(90.5) |
| | Yes | 27(9.3) | 28(9.7) | 55(9.5) |
| **Weight loss** | No | 266(91.7) | 272(93.8) | 538(92.8) |
| | Yes | 24(8.3) | 18(6.2) | 42(7.2) |
| **Night sweats** | No | 274(94.5) | 283(97.6) | 557(96.0) |
| | Yes | 16(5.5) | 7(2.4) | 23(4.0) |
| **Fever** | No | 269(92.8) | 276(95.2) | 545(94.0) |
| | Yes | 21(7.2) | 14(4.8) | 35(6.0) |

**Table 4. TB detected cases and associated factors in Amhara region.**

| Variables | Category | Active TB status | | OR | P value |
|---|---|---|---|---|---|
| | | Negative | Positive | | |
| **Gender** | Male | 273 | 5 | 1.36(0.36–5.13) | 0.744 |
| | Female | 298 | 4 | 1 | |
| **Age** | 18–24 | 68 | 0 | ND | |
| | 25–34 | 372 | 7 | ND | 0.480 |
| | 35–44 | 81 | 2 | ND | |
| | >44 | 50 | 0 | ND | |
| **Total years of experience** | <5 years | 309 | 5 | 1.31 (0.15–11.38) | 0.966 |
| | 5–10 years | 181 | 3 | 1.34(0.14–13.10) | |
| | >10 years | 81 | 1 | 1 | |
| **Trained on TB infection control** | No | 512 | 9 | - | 0.609 |
| | Yes | 59 | 0 | ND | |
| **BCG vaccinated** | No | 493 | 6 | 0.32(0.08–1.29) | 0.118 |
| | Yes | 78 | 3 | 1 | |
| **Worked on triaging for coughing patients** | No | 411 | 7 | 1 | 1.000 |
| | Yes | 160 | 2 | 0.73 (0.15–3.57) | |
| **Had direct contact with known TB patients** | Yes, in the health facility | 385 | 8 | 3.55 (0.44–28.63) | 0.387 |
| | Yes, out of health facility | 15 | 0 | - | |
| | No | 171 | 1 | 1 | |

ND: not determined

data-recording tools for documentation [24]. The findings from this study indicate that triaging was not implemented by 32% of the health facilities. Similarly, a study from South Africa documented that approximately half of the primary healthcare workers did not always triage patients with presumptive TB [25]. Knowledge gaps could be an underlying reason as close to 90% of the participants did not receive TB infection prevention training, and 74% of the studied health facilities did not have a TB prevention plan in place. Training of all staff is the facility management responsibility according to the Ethiopian TB guidelines [23]. Similarly, inadequate training was reported in Mozambique, in China and in South Africa [26–28]. This could contribute to poor implementation and practice of the tuberculosis prevention and control programs in the setting. In addition, for the quarter of health facilities that did have an IC plan; 70% of them did not audit their plan annually. However, monitoring of implementation, annual audit and timely feedback of health care practices should be performed to prevent and control TB transmission at the health care facility level as per the WHO 2019 TB guideline [24].

In this study, about 54% of the participants were working in poorly ventilated rooms. As part of the IC plan, facilities should assess the direction of airflow to reduce *M. tuberculosis* transmission to health workers, persons attending health care facilities or other persons in settings with a high risk of transmission [24, 29]. Tuberculosis prevalence among HCWs is associated with administrative control, environmental control measures, and personal protection [30]. In China, the absence of appropriate ventilation systems increased the risk of TB [31].

Limited availability of protective respirators for health care workers like observed in our study, where nearly 80% did not have adequate PPE, has also been documented in the Dominican Republic, Nepal and Mozambique [26, 32, 33]. Unavailability of protective equipment could further increase the risk of exposure to *M. tuberculosis* for health workers. Per the national TB guideline, healthcare workers should use respirators at work and infectious TB patients should use masks [23].

This study evaluated the healthcare workers active disease status using laboratory based molecular Xpert ®MTB/RIF method, which is a more sensitive method to detect active TB cases [34] then used in several other studies [35–37]. However, this study was not without limitations. Some data were collected by observation like the use of personnel protective equipment and ventilation status of rooms. Therefore, observer bias in the parts in evaluating the practices of prevention and control measures was controlled using the training of data collectors. In this study, we did not use additional investigations like cytology and x-ray for the diagnosis of TB as a high budget for would have been required, however, this means we may have missed some TB cases that were not symptomatic. TB screening was evaluated using self-reported symptoms only in this study although we used the more sensitive molecular Xpert ®MTB/RIF method to confirm TB. Moreover, rotation of staff through different facilities and areas/wards within facilities in terms of risk factors was not taken into account.

## Conclusions

The magnitude of active tuberculosis among healthcare workers was higher than in the general population. The limited implementation of infection control activities observed needs to be addressed to ensure proper containment of tuberculosis infection risks in the facilities.

## Acknowledgments

The authors thank the Tuberculosis Research Advisory Committee of the Ethiopian Federal Ministry of Health for providing the opportunity to conduct this study. The authors also acknowledge the cooperation of study participants, health facility staff and health facility administration to enable us to collect the data.

## Author Contributions

**Conceptualization:** Melashu Balew Shiferaw.

**Data curation:** Melashu Balew Shiferaw, Mulusew Alemneh Sinishaw, Desalegne Amare, Genetu Alem, Dawit Asefa, Eveline Klinkenberg.

**Formal analysis:** Melashu Balew Shiferaw, Mulusew Alemneh Sinishaw, Desalegne Amare, Genetu Alem, Dawit Asefa, Eveline Klinkenberg.

**Funding acquisition:** Melashu Balew Shiferaw.

**Investigation:** Melashu Balew Shiferaw, Mulusew Alemneh Sinishaw, Eveline Klinkenberg.

**Methodology:** Melashu Balew Shiferaw, Mulusew Alemneh Sinishaw, Desalegne Amare, Genetu Alem, Dawit Asefa, Eveline Klinkenberg.

**Project administration:** Melashu Balew Shiferaw.

**Resources:** Melashu Balew Shiferaw.

**Supervision:** Melashu Balew Shiferaw, Desalegne Amare, Genetu Alem, Dawit Asefa, Eveline Klinkenberg.

**Validation:** Melashu Balew Shiferaw, Eveline Klinkenberg.

**Visualization:** Melashu Balew Shiferaw.

**Writing – original draft:** Melashu Balew Shiferaw, Desalegne Amare, Eveline Klinkenberg.

**Writing – review & editing:** Melashu Balew Shiferaw, Mulusew Alemneh Sinishaw, Desalegne Amare, Genetu Alem, Dawit Asefa, Eveline Klinkenberg.

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
