## [Decision Letter · Decision Letter 0]

25 Feb 2021

PONE-D-21-03747

The magnitude of active tuberculosis disease among healthcare workers and support staff in healthcare settings of the Amhara region, Ethiopia

PLOS ONE

Dear Dr. Shiferaw,

Thank you for submitting your manuscript to PLOS ONE. After careful consideration, we feel that it has merit but does not fully meet PLOS ONE’s publication criteria as it currently stands. Therefore, we invite you to submit a revised version of the manuscript that addresses the points raised during the review process.

Please submit your revised manuscript. If you will need significantly more time to complete your revisions, please reply to this message or contact the journal office at plosone@plos.org. Please include the following items when submitting your revised manuscript:

We look forward to receiving your revised manuscript.

Kind regards,

Frederick Quinn

Academic Editor

PLOS ONE

Journal Requirements:

Reviewers' comments:

Reviewer's Responses to Questions

**Comments to the Author**

1. Is the manuscript technically sound, and do the data support the conclusions?

Reviewer #1: Yes

Reviewer #2: Yes

Reviewer #3: Yes

2. Has the statistical analysis been performed appropriately and rigorously? 

Reviewer #1: Yes

Reviewer #2: Yes

Reviewer #3: Yes

3. Have the authors made all data underlying the findings in their manuscript fully available?

Reviewer #1: No

Reviewer #2: Yes

Reviewer #3: Yes

4. Is the manuscript presented in an intelligible fashion and written in standard English?

Reviewer #1: Yes

Reviewer #2: Yes

Reviewer #3: Yes

5. Review Comments to the Author

Reviewer #1: The paper is of interest and well written. The discussion should be implemeented . Limits of the study should be better discussed. Minor spelling check are required. I suggest to accept after minor revision

Reviewer #2: Thank you for the opportunity to review this paper. TB among health care workers is an important issue and often under reported with huge implications for occupational safety and infection prevention and control. This is especially true now with COVID-19 as well.

I just have a few comments and suggestions for clarification.

1. May want to check grammar throughout especially plurals.

2. Page 8 lines 143-144 – The authors may want to clarify how ventilation was assessed? Was this simply whether windows were open or closed? Was there a window? Was ACH calculated?

What is meant by “ventilation in the room”? Which room? Did you assess infection control in all areas of all facilities? Or only in TB clinics/wards/rooms? Please clarify.

3. Page 10 line 195 – The authors mention that 78.7% did not have nor use N95 respirators. Can you specify what percentage did not have access to N95s versus percentage that did not use them?

4. Page 11 lines 216 – 218 – The authors state that the TB prevalence for HCWs is higher than the TB prevalence for Ethiopia (140 per 100,000) but do not state what the prevalence in HCWs is. Did the authors calculate a prevalence of TB among HCWs in this study for comparison? If so, may want to consider adding to the results section.

5. Table 2. What is meant by “room cleanness” and what impact does that have on airborne transmission and TB?

6. Table 4. Compare “had direct contact with known TB patients” in the table – but it is well known that contact with undiagnosed cases of TB is a greater risk than known TB patients on treatment. Did you assess the risk of contact with undiagnosed cases – perhaps in general medical wards or OPDs?

7. A general comment which may fit under limitations. In Ethiopia nursing staff rotates through different facilities and areas/wards within facilities at regular 3-6 month intervals. Did you consider this rotation of staff through the facility in terms of risk factors?

Reviewer #3: Comments.

The Title did not reflect the comparative nature of the work.in addition, I think prevalence should be used instead of magnitude of active tuberculosis.

Line 31: Please remove the 290 HCWs 290 Support staff as it has been captured in the result section to avoid repetition in the abstract section.

Line 39: The title of this manuscript did not reflect the comparative nature of the work between HCWs and Support Staff.

Line 40: TB infection prevention and control is the terminology to be used not TB infection prevention

Line 36: Repetition of study participants

Line 43-44: What is the magnitude of active TB in the general population?

Line 69: PPE abbreviation should be from line 69 at first mention

Line 101: Can the authors explain better how 2009 fiscal year will be captured in 2016/2017. This is rather confusing.

Methodology

Line 127: What inform the selection of 38 HFs? What is the average number of HFs in each of the selected zones?

Line 128: What inform the selection of 9 HCWs and 9suppor staff from facilities selected from the study? What is the staff strength of the HCWs and the Support Staff in each of the facility selected? Proportionate sampling should have been done to ensure adequate representation of each facility selected.

Line 143: How was the ventilation status of the rooms assessed? Which room was assessed for example, for the support staff? Were clinics assessed for the HCWS?

Line 145: Were trained professional nurses/ health officers’ part of the data collectors?

Lines 146-148: What happened to those who did not have any symptoms? All the participants should have been screened.

Line 205: What stage of the treatment were those five participants who were currently on treatment because for them to still be positive

Line 222: TBIC should be written in full at the first mention

References:

Line 307: date the reference cited should be included

Line 312: Page numbers should be provided

Results: There is no need to put study date on the title

Table 1: some of the variables in some cell is more than 100%

It is better to list all the support staff before HCWs.

What is the difference between case managers and treatment supporters?

The statistical test employed and the p-value that informed the comparison made in the text that showed that HCWs and Support staff did not differ significantly was not presented in the tables.

Grid lines should be removed from the tables.

In addition, there were lots of grammatical errors and syntax in the manuscript, hence the work should be edited for example but not limited to the following:

Line 26…” can be utilized”

Line 30: “to recruit” would have been better than “to target”

Line 73: ………….knowing the prevalence of active TB among HCWs

Line 81: besides ‘the’ use of PPE

Line 112: “Cashiers” not “Cashers”.

Line 162: “lab” should read “laboratory personnel”

Line 177: male/female “ratio” not “ration”

Line 197: “of whom” should read “of which”

6. PLOS authors have the option to publish the peer review history of their article (what does this mean?). If published, this will include your full peer review and any attached files.

Reviewer #1: No

Reviewer #2: No

Reviewer #3: No

---

## [Author Response · Author response to Decision Letter 0]

2 Mar 2021

Dear Editor, 

Thank you for giving us the opportunity to revise our manuscript. We also thank the reviewers for their critical review of our manuscript and providing us important comments that help us to significantly improve our paper in advance. The following is our point by point response. 

Journal Requirements:

Response: thank you we checked the references and revised as seen in the updated version of the manuscript.

5. Review Comments to the Author

Reviewer #1: The paper is of interest and well written. The discussion should be implemented . Limits of the study should be better discussed. Minor spelling check are required. I suggest to accept after minor revision

Response: Thank you for the comments. Now, the comments are accepted and have been addressed in the updated manuscript 

Reviewer #2: Thank you for the opportunity to review this paper. TB among health care workers is an important issue and often under reported with huge implications for occupational safety and infection prevention and control. This is especially true now with COVID-19 as well.

I just have a few comments and suggestions for clarification.

1. May want to check grammar throughout especially plurals.

Response: thank you for the comment, and it has been addressed in the new version manuscript

2. Page 8 lines 143-144 – The authors may want to clarify how ventilation was assessed? Was this simply whether windows were open or closed? Was there a window? Was ACH calculated?

Response: thank you, the ventilation of the room was evaluated and considered ventilated if a room had open window and cross ventilation was present with the door. ACH was not calculated. We added a better definition of our measurements in the new version manuscript

What is meant by “ventilation in the room”? Which room? Did you assess infection control in all areas of all facilities? Or only in TB clinics/wards/rooms? Please clarify.

Response: thank you, how we measured room ventilation is outline above. As presented in table two, 381 of the participants (91 support staff and 290 HCWs) worked in different rooms such as TB clinic, card room, x-ray, laboratory, etc. Of which, 174 participants worked in well ventilated rooms in the health facilities of the Amhara region, Ethiopia (Table 2). Others who had no room to work in like security guards did not have rooms to be evaluated. We made this now more clear in the method section of the manuscript. We did assess infection control in all areas of facilities included in the study this is outlined in in the updated manuscript. 

3. Page 10 line 195 – The authors mention that 78.7% did not have nor use N95 respirators. Can you specify what percentage did not have access to N95s versus percentage that did not use them?

Response: thank you for the comments. Of the total 124 participants who had N95 respirator access, 82 (66.1%) did not use the respirator to prevent TB infection. This has now been made more explicit in the results section of the revised paper. 

4. Page 11 lines 216 – 218 – The authors state that the TB prevalence for HCWs is higher than the TB prevalence for Ethiopia (140 per 100,000) but do not state what the prevalence in HCWs is. Did the authors calculate a prevalence of TB among HCWs in this study for comparison? If so, may want to consider adding to the results section.

Response: The prevalence of TB disease found in this study was (1.6% ~ 1600 per 100000 populations). Now, it has been included as per the reviewer’s recommendation 

5. Table 2. What is meant by “room cleanness” and what impact does that have on airborne transmission and TB?

Response: thank you for the comment. Room cleanness was evaluated by interview and observation of the room using a standardize checklist. It was included to check the implementation of general infection prevention practices like cleaning of the floor, benches, chairs, etc with appropriate detergent such as bleach solution. 

6. Table 4. Compare “had direct contact with known TB patients” in the table – but it is well known that contact with undiagnosed cases of TB is a greater risk than known TB patients on treatment. Did you assess the risk of contact with undiagnosed cases – perhaps in general medical wards or OPDs?

Response: thank you for this constructive comment. Honestly, we did not assess the risk of contact with undiagnosed cases assuming all of the HCWs and support staff would have had a risk of contact then.

7. A general comment which may fit under limitations. In Ethiopia nursing staff rotates through different facilities and areas/wards within facilities at regular 3-6 month intervals. Did you consider this rotation of staff through the facility in terms of risk factors?

Response: thank you for the comment. We did not consider rotation of staff through different facilities and areas/wards within facilities in terms of risk factors. We have now included this issue under the limitations in the updated version manuscript.

Reviewer #3: Comments.

The Title did not reflect the comparative nature of the work. In addition, I think prevalence should be used instead of magnitude of active tuberculosis.

Response: thank you, now the title has been modified in the new version manuscript. 

Line 31: Please remove the 290 HCWs 290 Support staff as it has been captured in the result section to avoid repetition in the abstract section.

Response: thank you for this comment. Now, it has been removed from the abstract section

Line 39: The title of this manuscript did not reflect the comparative nature of the work between HCWs and Support Staff.

Response: thank you for the comments, our objective is to show the burden of active tuberculosis equally affected support staff as healthcare workers. The prevalence found in HCWs is 1.4 and in support staff 1.7% but the difference was not significant indicating both of them equally suffered from TB diseases

Line 40: TB infection prevention and control is the terminology to be used not TB infection prevention

Response: thank you, now it has been corrected

Line 36: Repetition of study participants

Response: thank you, now it has been corrected

Line 43-44: What is the magnitude of active TB in the general population?

Response: thank you, in the general population, it is 140 per 100000 populations. And now it has been included in the new version manuscript.

Line 69: PPE abbreviation should be from line 69 at first mention

Response: thank you, now it has been abbreviated as the reviewer’s advice

Line 101: Can the authors explain better how 2009 fiscal year will be captured in 2016/2017. This is rather confusing.

Response: thank you, the 2009 Ethiopian fiscal year is 08 July 2016 to 07 July 2017

Methodology

Line 127: What inform the selection of 38 HFs? What is the average number of HFs in each of the selected zones?

The average number of health facilities (HFs) in each selected zone was 52 in Awi, 12 in Bahir Dar, 102 in South Gondar, 109 in East Gojjam, and 70 in North Wollo. Then, proportionally we allocated 6 HFs from Awi, 2 HFs from Bahir Dar, 11 HFs from South Gondar, 12 HFs from East Gojjam, and 7 HFs from North Wollo zones making a total of 38 sampled HFs. More details on the selection process has been added in the updated version manuscript. 

Line 128: What inform the selection of 9 HCWs and 9suppor staff from facilities selected from the study? What is the staff strength of the HCWs and the Support Staff in each of the facility selected? Proportionate sampling should have been done to ensure adequate representation of each facility selected.

Response: thank you for this comment really help to improve our manuscript. This was planned to represent 580 samples equally distributed to the 38 health facilities making 16 participants per site (580/38=15.3~16). That is 8 HCWs and 8 support staff interviewed per facility. The 9 HCWs and 9 support staff described was clerical error and now it has been corrected. 

Line 143: How was the ventilation status of the rooms assessed? Which room was assessed for example, for the support staff? Were clinics assessed for the HCWS?

Response: The ventilation of the room was evaluated as well ventilated if a room had open window and cross ventilated with the door. As presented in table two, 381 of the participants (91 support staff and 290 HCWs) worked in different rooms such as TB clinic, card room, x-ray, laboratory, etc. Of which, 174 participants worked in well ventilated rooms in the health facilities of the Amhara region, Ethiopia (Table 2). Others who had no room to work in like security guards did not have rooms so that not evaluated. 

Line 145: Were trained professional nurses/ health officers’ part of the data collectors?

Response: thank you for the comments. Yes, first we trained professional nurses/health officers on how to fill the questionnaire and the way on how to collect during observation. Then, those trained data collectors evaluated TB symptoms from the study participants. 

Lines 146-148: What happened to those who did not have any symptoms? All the participants should have been screened.

Response: thank you for the constructive comments. Based on the Ethiopian national TB screening algorithm, those who had no any active TB symptoms were considered as TB screening negative. This has been made more explicit in the revised manuscript 

Line 205: What stage of the treatment were those five participants who were currently on treatment because for them to still be positive

Response: thank you, those five patients were on the intensive phase of TB treatment, and now we describe in the updated manuscript 

Line 222: TBIC should be written in full at the first mention

Response: thank you, TBIC refers to TB infection control, and now it has been written in full at the first mention in the new version manuscript

References:

Line 307: date the reference cited should be included

Response: thank you, now date the reference cited has been included

Line 312: Page numbers should be provided

Response: Thank you, now it has been provided in the new version manuscript 

Results: There is no need to put study date on the title

Response: thank you, study date has been removed on the title of result tables

Table 1: some of the variables in some cell is more than 100%

It is better to list all the support staff before HCWs.

Response: thank you, the variables having more than 100% has been revised, and support staff has been listed before HCWs. See updated version manuscript

What is the difference between case managers and treatment supporters?

Response: Thank you, Case manager: support staff assigned to help patients in guiding and facilitating the diagnosis and treatment in health facilities; Treatment supporter: support staff assigned to follow treatment adherence of patients using different communication mechanisms such as phone call. We have now described this better in the revised version of the manuscript 

The statistical test employed and the p-value that informed the comparison made in the text that showed that HCWs and Support staff did not differ significantly was not presented in the tables.

Response: thank you, p value has been presented in the table 2 of the updated manuscript. 

Grid lines should be removed from the tables.

Response: thank you, the grid lines has been removed from the tables in the updated manuscript

In addition, there were lots of grammatical errors and syntax in the manuscript, hence the work should be edited for example but not limited to the following:

Line 26…” can be utilized”

Response: thank you, now it has been revised

Line 30: “to recruit” would have been better than “to target”

Response: thank you, now it has been revised

Line 73: ………….knowing the prevalence of active TB among HCWs

Response: thank you, now it has been revised

Line 81: besides ‘the’ use of PPE

Response: thank you, now ‘the’ has been included

Line 112: “Cashiers” not “Cashers”.

Response: thank you, now it has been corrected

Line 162: “lab” should read “laboratory personnel”

Response: thank you, now it has been corrected

Line 177: male/female “ratio” not “ration”

Response: thank you, now it has been corrected

Line 197: “of whom” should read “of which”

Response: thank you, now it has been revised

---

## [Decision Letter · Decision Letter 1]

26 Mar 2021

PONE-D-21-03747R1

Prevalence of active tuberculosis disease among healthcare workers and support staff in healthcare settings of the Amhara region, Ethiopia

PLOS ONE

Dear Dr. Shiferaw,

Thank you for submitting your manuscript to PLOS ONE. After careful consideration, we feel that it has merit but does not fully meet PLOS ONE’s publication criteria as it currently stands. Therefore, we invite you to submit a revised version of the manuscript that addresses the points raised during the review process.

Please submit your revised manuscript. If you will need significantly more time than this to complete your revisions, please reply to this message or contact the journal office at plosone@plos.org. Please include the following items when submitting your revised manuscript:

We look forward to receiving your revised manuscript.

Kind regards,

Frederick Quinn

Academic Editor

PLOS ONE

Journal Requirements:

Reviewers' comments:

Reviewer's Responses to Questions

**Comments to the Author**

1. If the authors have adequately addressed your comments raised in a previous round of review and you feel that this manuscript is now acceptable for publication, you may indicate that here to bypass the “Comments to the Author” section, enter your conflict of interest statement in the “Confidential to Editor” section, and submit your "Accept" recommendation.

Reviewer #1: All comments have been addressed

Reviewer #2: All comments have been addressed

Reviewer #3: All comments have been addressed

2. Is the manuscript technically sound, and do the data support the conclusions?

Reviewer #1: Yes

Reviewer #2: Yes

Reviewer #3: Yes

3. Has the statistical analysis been performed appropriately and rigorously? 

Reviewer #1: Yes

Reviewer #2: Yes

Reviewer #3: Yes

4. Have the authors made all data underlying the findings in their manuscript fully available?

Reviewer #1: No

Reviewer #2: Yes

Reviewer #3: Yes

5. Is the manuscript presented in an intelligible fashion and written in standard English?

Reviewer #1: Yes

Reviewer #2: Yes

Reviewer #3: Yes

6. Review Comments to the Author

Reviewer #1: The manuscript was substantially implemented. All comments have been addressed. The paper can be accepted for publication.

Reviewer #2: (No Response)

Reviewer #3: The authors had done the corrections but there are still some grammatical errors. Other comments have been attached

7. PLOS authors have the option to publish the peer review history of their article (what does this mean?). If published, this will include your full peer review and any attached files.

Reviewer #1: No

Reviewer #2: No

Reviewer #3: No

---

## [Author Response · Author response to Decision Letter 1]

10 May 2021

Dear editor, thank you for giving us the opportunity to revise our manuscript and we also thank the reviewers for comments and critics helping us to improve our paper. The following are point by point responses.

The authors had done the corrections but the manuscript still need to be edited for grammatical errors. 

Response: thank you, grammatical errors have been corrected throughout the whole manuscript. 

In addition, I suggest that the variables in table one should be listed in descending other;

Support staff: 

Cleaner 58(20.0)

Data clerk 51(17.6)

Cashier 42(14.5)

HCWs:

Nurse 122 (42.1)

Laboratory staff 50(17.2)

Pharmacy staff 34(11.7)

Response: thank you for the valuable comment. Now, table one has been updated according to the reviewer’s advice

---

## [Decision Letter · Decision Letter 2]

31 May 2021

Prevalence of active tuberculosis disease among healthcare workers and support staff in healthcare settings of the Amhara region, Ethiopia

PONE-D-21-03747R2

Dear Dr. Shiferaw,

We’re pleased to inform you that your manuscript has been judged scientifically suitable for publication and will be formally accepted for publication once it meets all outstanding technical requirements.

Kind regards,

Frederick Quinn

Academic Editor

PLOS ONE

Additional Editor Comments (optional):

Reviewers' comments:

Reviewer's Responses to Questions

**Comments to the Author**

1. If the authors have adequately addressed your comments raised in a previous round of review and you feel that this manuscript is now acceptable for publication, you may indicate that here to bypass the “Comments to the Author” section, enter your conflict of interest statement in the “Confidential to Editor” section, and submit your "Accept" recommendation.

Reviewer #2: All comments have been addressed

Reviewer #3: All comments have been addressed

2. Is the manuscript technically sound, and do the data support the conclusions?

Reviewer #2: Yes

Reviewer #3: Yes

3. Has the statistical analysis been performed appropriately and rigorously? 

Reviewer #2: Yes

Reviewer #3: Yes

4. Have the authors made all data underlying the findings in their manuscript fully available?

Reviewer #2: Yes

Reviewer #3: (No Response)

5. Is the manuscript presented in an intelligible fashion and written in standard English?

Reviewer #2: No

Reviewer #3: Yes

6. Review Comments to the Author

Reviewer #2: Please correct the grammar and language issues as noted in the original review. It still appears that these have not been corrected.

Reviewer #3: Thanks for asking me to review the manuscript titled: “The Prevalence of active tuberculosis disease among healthcare workers and support staff in healthcare settings of the Amhara region, Ethiopia.” (PONE-D-21-03747R2) for the third time.

The authors had done the corrections and the manuscript has been edited for grammatical errors.

7. PLOS authors have the option to publish the peer review history of their article (what does this mean?). If published, this will include your full peer review and any attached files.

Reviewer #2: No

Reviewer #3: No

---

## [Editor Report · Acceptance letter]

3 Jun 2021

PONE-D-21-03747R2 

Prevalence of active tuberculosis disease among healthcare workers and support staff in healthcare settings of the Amhara region, Ethiopia 

Dear Dr. Shiferaw:

I'm pleased to inform you that your manuscript has been deemed suitable for publication in PLOS ONE. Congratulations! Your manuscript is now with our production department. 

Kind regards, 

on behalf of

Dr. Frederick Quinn 

Academic Editor

PLOS ONE